# SARS-CoV-2 detection and genomic sequencing from hospital surface samples collected at UC Davis

David A. Coil[1], Timothy Albertson[2], Shefali Banerjee[3], Greg Brennan[3], A. J. Campbell[4], Stuart H. Cohen[5], Satya Dandekar[3], Samuel L. Díaz-Muñoz[1,4], Jonathan A. Eisen[1,3,6], Tracey Goldstein[7], Ivy R. Jose[4], Maya Juarez[2], Brandt A. Robinson[2], Stefan Rothenburg[3], Christian Sandrock[2], Ana M. M. Stoian[3], Daniel G. Tompkins[2], Alexandre Tremeau-Bravard[7], Angela Haczku[2]*

1 Genome Center, University of California, Davis, California, United States of America, 2 Division of Pulmonary, Critical Care and Sleep Medicine, Department of Internal Medicine, School of Medicine, University of California, Davis, California, United States of America, 3 Department of Medical Microbiology and Immunology, School of Medicine, University of California, Davis, California, United States of America, 4 Department of Microbiology and Molecular Genetics, College of Biological Sciences, University of California, Davis, California, United States of America, 5 Division of Infectious Diseases, Department of Internal Medicine, School of Medicine, University of California, Davis, California, United States of America, 6 Department of Evolution and Ecology, University of California, Davis, California, United States of America, 7 One Health Institute, University of California, Davis, California, United States of America

* haczku@ucdavis.edu

**Data Availability Statement:** All relevant data are uploaded to GitHub (https://github.com/sociovirology/sars_cov2_environmental_seq).

## Abstract

### Rationale

There is little doubt that aerosols play a major role in the transmission of SARS-CoV-2. The significance of the presence and infectivity of this virus on environmental surfaces, especially in a hospital setting, remains less clear.

### Objectives

We aimed to analyze surface swabs for SARS-CoV-2 RNA and infectivity, and to determine their suitability for sequence analysis.

### Methods

Samples were collected during two waves of COVID-19 at the University of California, Davis Medical Center, in COVID-19 patient serving and staff congregation areas. qRT-PCR positive samples were investigated in Vero cell cultures for cytopathic effects and phylogenetically assessed by whole genome sequencing.

### Measurements and main results

Improved cleaning and patient management practices between April and August 2020 were associated with a substantial reduction of SARS-CoV-2 qRT-PCR positivity (from 11% to 2%) in hospital surface samples. Even though we recovered near-complete genome sequences in some, none of the positive samples (11 of 224 total) caused cytopathic effects

**Funding:** Funding for this work was provided by a UC Davis CRAFT Award (JAE and DAC) and The Chester Robbins Endowment (AH).

**Competing interests:** The authors have declared that no competing interests exist.

in cultured cells suggesting this nucleic acid was either not associated with intact virions, or they were present in insufficient numbers for infectivity. Phylogenetic analysis suggested that the SARS-CoV-2 genomes of the positive samples were derived from hospitalized patients. Genomic sequences isolated from qRT-PCR negative samples indicate a superior sensitivity of viral detection by sequencing.

## Conclusions

This study confirms the low likelihood that SARS-CoV-2 contamination on hospital surfaces contains infectious virus, disputing the importance of fomites in COVID-19 transmission. Ours is the first report on recovering near-complete SARS-CoV-2 genome sequences directly from environmental surface swabs.

## Introduction

There is a paucity of data regarding survival and infectivity of the SARS-CoV-2 virus on surfaces in closed environments, although some data are available for other coronaviruses [1, 2]. Early in the pandemic, testing of artificially generated aerosols on copper, stainless steel, cardboard, and plastic surfaces found a rapid decay of viral viability within a few days [3]. Another study examining survival on PPE showed that the virus decayed rapidly on cotton but survived for up to 21 days on some other surface material [4]. More recent evaluation of a variety of surfaces showed that infectious virions could survive for up to 28 days in laboratory conditions including high titer virus and in the dark [5]. However, it is unclear in all of these cases how this relates to virus survival and the potential for its transmission outside the laboratory. A study of high-touch surfaces in a community setting attempted to estimate transmission risk, but there are still too many unknowns to do this with any confidence [6]. It is known that SARS-CoV-2 can survive on skin for about nine hours and may allow or extend viral survival on surfaces following contact [7].

A key complication in studies of SARS-CoV-2 environmental viability relates to how long the viral RNA can be detected on surfaces. A large number of studies have used qRT-PCR to detect SARS-CoV-2 viral RNA indoors [8–20] reviewed in [21] and found that the virus was detectable up to several weeks after it was presumably deposited [22]. The amount of viral RNA detected seems to be inversely correlated with cleaning protocols [23]. This probably explains otherwise surprising results such as the lack of viral RNA detected in an oncology ward housing patients with COVID-19 [24], or the very low probability of detection in an ICU [25]. Several studies detected SARS-CoV-2 RNA in these environments but were unable to culture infectious SARS-CoV-2 virions [26–28]. However, viable SARS-CoV-2 was successfully cultured and sequenced from the air of the hospital room with a COVID-19 patient using a water vapor condensation method [29].

In this study, we assessed environmental contamination with SARS-CoV-2 in a hospital setting by both qRT-PCR and a viral culture assay. We examined surfaces, and also sampled HVAC filters since these have been previously shown to contain SARS-CoV-2 in healthcare settings [30, 31] and in homes [22]. In addition, we sequenced partial and complete genomes from surfaces and compared them phylogenetically to identify the source of the virus.

## Materials and methods

### Swab sample collection at the UC Davis Medical Center (UCDMC)

UCDMC is a 625-bed academic medical center in Northern California. While there are multiple ICUs and medical floors, during the first 6 months of the pandemic, most patients with active COVID-19 were hospitalized in 3 intensive care units (ICU) and 2 medical wards. Both the ICU and medical wards have the ability to place individual rooms as well as the entire ward under negative pressure, and that was the case during the study. Samples were collected using standard Puritan cotton-tipped swabs with plastic handles and placed into Trizol as described below. The first set of samples was collected in April 2020, and the second set between late July/early August 2020. Clinical staff swabbed an approximately 10cm x 10 cm area for several seconds, as if trying to clean it with a scrubbing motion and rotating the swab.

**Heating, Ventilation, and Air Conditioning (HVAC) swab collection.** Swabs were moistened in saline, brushed across the air filters, and then placed into 500 ul of Trizol(R). For safety reasons, the air pressure in the HVAC system was temporarily reduced during sampling. Sampling took place on the filters which protect the evaporator coils from dust, meaning that the sampled dust was unfiltered directly from the hospital floor. Samples were collected both from the floor with a number of COVID-19 patients, as well as from another floor with no known COVID-19 patients. All samples were frozen at -80 ˚C until processing.

**Surface sampling.** During the first collection, swabs were pre-moistened in sterile saline and then placed into 500 uL Trizol(R); during the second round, swabs were either pre-moistened with Trizol(R) or viral transport media (VTM, Innovative Research™) and then placed into their respective individual containers after sample collection. All samples were stored frozen at -80 ˚C until processing.

**Surface sampling (for viability testing).** For viability testing, a pair of swabs were held together for the swabbing. One was placed in Trizol for qRT-PCR (as described above) and the other into VTM. All samples were stored frozen at -80 ˚C until processing.

### qRT-PCR

RNA extraction from swabs was performed using the Zymo Research Direct-zol-96RNA kit (#R2054). Briefly, 500 ul of pure ethanol was added to the 500 ul of Trizol+swab. The mixture was transferred to a I-96 plate extraction performed according to the manufacturer instructions. RNA was eluted in 25 ul water and cDNA was made using the SuperScriptIII ThermoFisher kit (#18080051). SARS-CoV-2 screening was performed by qRT-PCR using Taqman Universal Master Mix II+UNG (ThermoFisher #4440038). Primers and probes and cycling conditions to detect segments of the N and RdRp genes were performed following the CDC (https://www.cdc.gov/coronavirus/2019-ncov/lab/rt-pcr-panel-primer-probes.html) and Corman et al. protocols [32]. qRT-PCR was run for 45 cycles and any positive signal was reported.

### Vero cell culture and SARS-CoV-2 infection studies

Vero E6 cells (ATCC #CRL-1586) were maintained in Dulbecco's Modified Eagle's Medium (DMEM) supplemented with 10% fetal bovine serum (FBS) and 100 IU/ml of penicillin-streptomycin (Pen-Strep; Gibco). The mNeonGreen SARS-CoV-2 (icSARS-CoV-2-mNG) virus [33] was kindly provided by the UTMB World Reference Center for Emerging Viruses and Arboviruses and Dr. Scott Weaver, and was propagated and titered in Vero E6 cells. All swab samples and positive controls were diluted in D10-CoV medium consisting of DMEM supplemented with 10% FBS, 100 IU/ml Pen-Strep, 250 μg/ml Amphotericin B (Gibco) and 250 μg/ml Gentamicin (Quality Biologicals).

Six-well plates of Vero E6 cells (~60% confluent) were infected with either 300 uL of the viral transport medium from qRT-PCR positive environmental swab samples diluted 1:1 in D10-CoV medium, or 300 μL of mNeonGreen SARS-CoV-2 (icSARS-CoV-2-mNG) 10-fold serially diluted in D10-CoV medium to infect wells with $10^5$ PFU to $10^0$ PFU per well. Following 1h incubation at 37 ˚C, rocking plates every 15 minutes, the cells were replenished with fresh D10-CoV medium and incubated at 37 ˚C + 5% $CO_2$ for five days. A mock-treated control consisting of cells only maintained in D10-CoV medium was included in the assay and treated identically. All samples were tested in duplicate. Two and five days post-infection, the cells were assessed microscopically for any visible cytopathic effect. Five days post infection, 2 mL of cell culture supernatant was collected from each well and mixed with 6 mL of Trizol LS reagent (Ambion). Cell lysates were harvested by adding 1 mL of Trizol LS reagent to the cell monolayer. All Trizol-treated samples were used for RNA extraction and qRT-PCR.

### SARS-CoV-2 viral genome sequencing

We prepared RNA extractions for Oxford Nanopore (ONT) MinION sequencing of SARS-CoV-2 viral genomes. We made modifications to the ARTIC Network Protocol (v2) [34], to optimize sequencing of environmental samples. Our complete protocol is available online https://www.protocols.io/view/ncov-2019-environmental-sample-sequencing-protocol-brnbm5an. In brief: we conducted random hexamer primed reverse transcription and amplified cDNA using v3 primers, which tile the entire viral genome (save for non-coding regions at the genome ends) with overlapping 400 bp fragments. We concentrated PCR products using the Zymo Select-a-Size DNA Clean & Concentrator Kit (Zymo Research, Irvine CA), ligated barcodes using the Oxford Nanopore Native Barcoding kit, and ligated sequencing adaptors. Samples were run on ONT R9.4 or R10.3 flow cells. We followed the ARTIC Network bioinformatics SOP, which in brief involved high accuracy basecalling and demultiplexing using ONT Guppy, mapping reads to the Wuhan-Hu-1 (accession MN908947) reference, polishing with Nanopolish, and consensus generation (code for analysis available https://github.com/sociovirology/sars_cov2_environmental_seq).

## Results and discussion

### Improved cleaning protocol and patient management was associated with decreased recovery of SARS-CoV-2 RNA from hospital surface samples

During the first wave of COVID-19 (March-April, 2020) the role of fomites in transmission was controversial and studies providing supporting evidence for it were lacking. Some of our hospital personnel also became ill with COVD-19 at that time. To investigate whether the infection clusters among health care workers were associated with SARS-CoV-2 contaminated areas, we collected 56 swabs in April 2020, from a variety of frequently used locations. Six of these samples (11%) tested positive for SARS-CoV-2 by qRT-PCR for the viral N1 and N2 genes (Fig 1). While the positive locations were in the proximity of hospitalized COVID-19 patients, none of these areas were related to where the hospital personnel cluster infections were suspected to originate from.

During a three-month period between April and August 2020, important changes took place to improve cleaning protocols with a change in the frequency/duration/composition of cleaning material in the hospital. In addition to the cleaning protocol changes, improved patient management of respiratory secretions took place. This included earlier intubation, rapid sequence ventilation, and changes in the management of high $O_2$ flow nasal cannulas. To investigate whether changes in cleaning practices and patient management impacted the

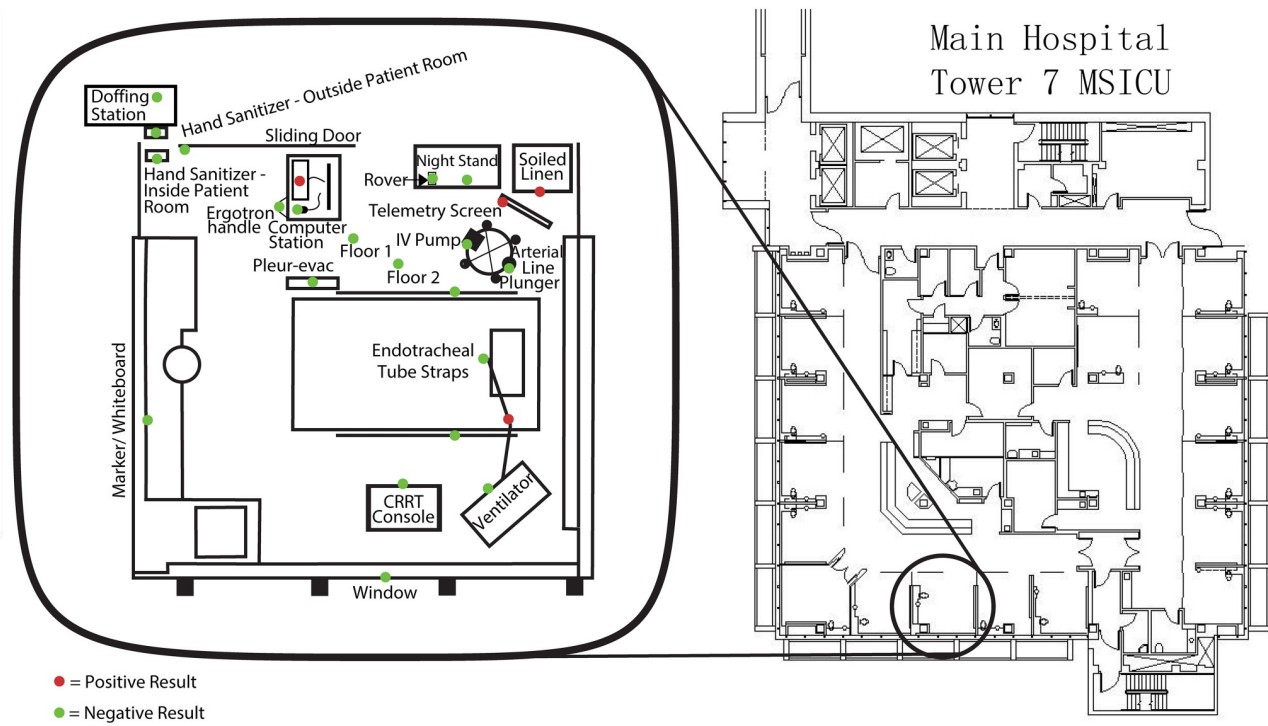

**Fig 1. Representative ward room sampled by swabs for SARS-CoV-2 at the UCDMC.** Positive samples are shown in red, negative samples in green. Each dot represents a single swab.

outcomes compared to our earlier findings, we performed a follow-up study by collecting an additional 168 swabs. Out of these, only five tested positive for SARS-CoV-2 by qRT-PCR (S1 and S2 Tables). None of the HVAC samples were positive by qRT-PCR.

Thus, our results show a substantial decrease in positive samples from 11% to 2% between April and August. This trend is particularly significant in the light that in mid-August, 2020, a second surge of COVID-19 cases were admitted, substantially increasing the number of patients in the hospital (Fig 2).

We propose that together, the improved cleaning protocols and patient management practices likely contributed to decreased presence of aerosolized (and deposited) virions in the

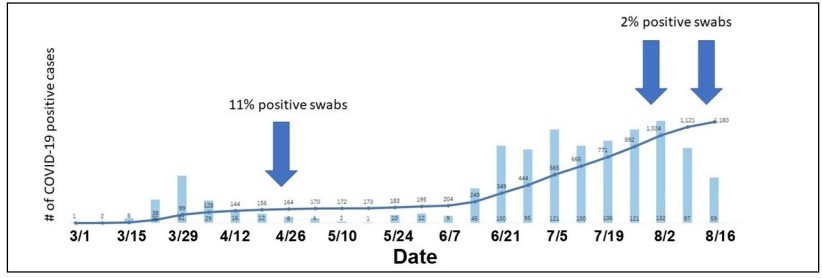

**Fig 2. SARS-CoV-2 positive patients at UCDMC during the first and the second wave of COVID-19.** Weekly totals of COVID-19 patients, and the cumulative total number from early March until mid-August, 2020. The blue arrows indicate the sampling dates.

rooms where COVID-19 patients were cared for. It was still unclear however, whether the recovered viral RNA from the samples collected from hospital surfaces could be a feasible source of infection.

## Hospital surface SARS-CoV-2 RNA did not exhibit infectious nature in a Vero cell culture model *in vitro*

To investigate whether the SARS-CoV-2 qRT-PCR positivity in hospital surface samples was associated with potential infectivity, a total of five swabs (identified as positive by qRT-PCR) were tested. We used an *in vitro* infection assay to detect the presence of infectious virus particles. Each of the wells of Vero E6 cells incubated with individual swab samples appeared identical to the mock-infected cells and showed no signs of cytopathic effect (CPE) by microscopy for up to five days post-infection (dpi) (Fig 3). This lack of CPE in swab-inoculated wells was consistent in two biologically independent infection assays in all tested samples. In contrast, positive control samples infected with 10-fold serial-dilutions from $10^5$ to 1 PFU of mNeon-Green SARS-CoV-2 showed notable CPE and mNeonGreen expression throughout the course of infection, even in wells infected with only 1 PFU (Fig 3). Therefore, the lack of CPE in the environmental swab samples indicated the absence of infectious virus particles or samples with a viral load below the detection limit for viral culture.

To confirm this result, supernatant and cell lysates from the swab and positive control inoculated Vero E6 cells were collected five dpi from each independent experiment. Total RNA from each sample was analyzed by qRT-PCR assay in duplicate, and while no signal was observed with the N1 primer set, a low signal (CT 28, 37) was detected in two of the samples with the N2 primer set. A repeat of this experiment in triplicate for each sample only yielded low signal in a single reaction (CT 37). In combination with the lack of viral infectivity in cell culture assays, our data suggest that the signal most likely represented relic RNA from the original swab and not due to the replication of viral particles in culture.

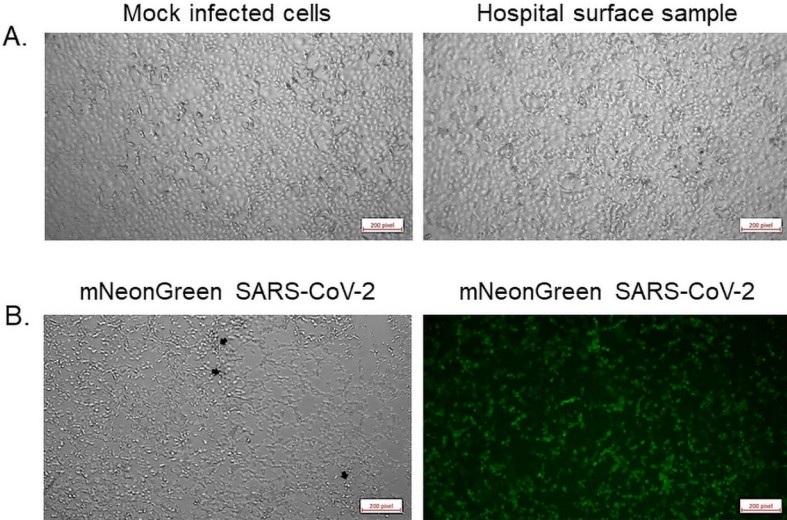

**Fig 3. Micrographs of Vero E6 cells five days after inoculation.** Cells were either mock-infected (upper left), inoculated with swab samples (representative of all five tested samples, upper right), or infected with one PFU of mNeonGreen SARS-CoV-2 (phase contrast, lower left; mNeonGreen lower right).

## Viral genome sequencing

In order to determine the genome sequences from the isolated samples, we generated a total of 17,567,849 reads across five separate MinION sequencing runs (S3 Table), of which 6,670,616 were used for mapping after demultiplexing and quality control. The negative control in Run 4 yielded reads that mapped to the reference genome, therefore samples were re-sequenced in Run 5. Negative controls in Runs 1–3 and 5 had no reads mapping to the reference genome. At least one positive control (included in Runs 4 and 5), per run produced reads that mapped to the reference genome (details in GitHub repository https://github.com/sociovirology/sars_cov2_environmental_seq).

The genome coverage obtained from samples was assigned to three groups: >15% (n = 61), 20–40% (n = 5), >75% (n = 5). The percent of the genome covered at a 5X depth quickly declined as a function of increasing mean Ct values (Fig 4). There was a notable threshold of Ct ~ 38, above which no sample achieved >10% genome completeness.

## Whole-genome PCR and sequencing yields more effective detection of SARS-CoV-2 than qRT-PCR

While there was a steep drop-off in achieving a full genome sequence with increasing Ct values, the sequencing protocol was able to detect SARS-CoV-2 in samples with undetermined Ct scores by PCR, with an average of 6.27% coverage (range: 2.19–14.78%). Using a sequencing cutoff of >2% genome coverage, sequences of SARS-CoV-2 were amplified in 15 samples that had no detectable Ct by PCR, whereas five samples that did not have a detectable Ct were not amplified by sequencing (at >2% coverage). This uncoupling of detection by qRT-PCR vs sequencing is likely due to the fact that qRT-PCR targets only a small portion of the genome and sequencing primers cover the entire genome (e.g. [35]). Furthemore, environmental

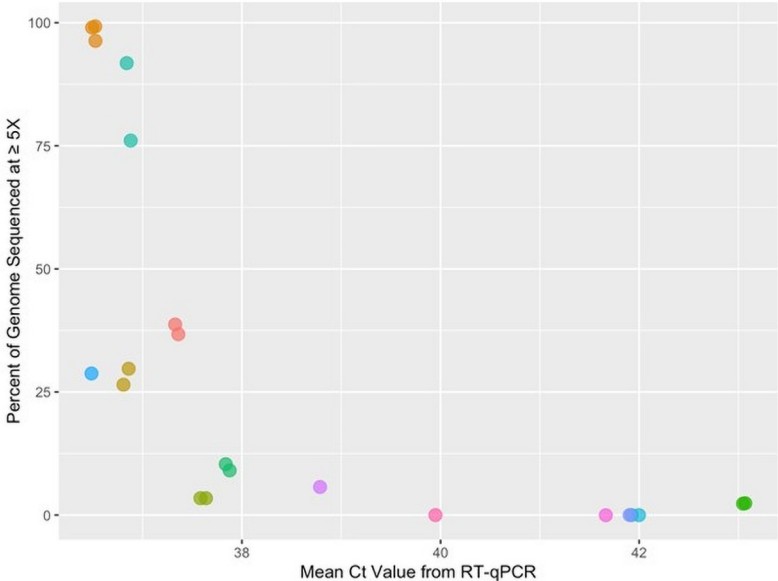

**Fig 4. Environmental swabs with Ct values below 38 yielded enough sequence reads to cover a substantial portion of the SARS-CoV-2 genome.** The percent of the SARS-CoV-2 reference genome (isolate Wuhan-Hu-1) covered at ≥ 5X decreased steeply as a function of the mean Ct value (using CDC N1, N2, and Berlin RdRP primers). The colored points represent individual swab samples, some of which were re-run in independent sequencing runs.

samples in particular may have been degraded or diluted, affecting the genomic RNA available for reverse transcription, as observed in multiple studies of environmental samples [36–38].

## Generation of near-complete genomes from environmental samples

We recovered two near-complete genomes from two different patient rooms, D14 and T7 Blue. These samples were collected from two surfaces, the floor and a soiled linens basket lid. Genome coverage and Ct values for D14 were 99.26% (Mean Ct = 36.49) and T7 Blue 91.75% (Mean Ct = 36.89), both with a depth cutoff of 5X to call a base. The sample from room D14 had an average depth of 371.21 ± 171.30 reads (mean ± SD). The sample from room T7 had an average depth of 377.14 ± 185.03.

## Effect of protocol modifications for environmental sample sequencing

The ARTIC protocol was modified in two major ways to accommodate the lower sample concentration in environmental samples compared to clinical samples: concentration and cleaning of PCR products and making duplicate barcoding reactions. Concentration of PCR products increased the genome coverage from 96.31% to 99.02% (sample from room D14) and from 76.08% to 91.75% (sample from room T7 Blue), compared to the standard ARTIC protocol. Duplicate barcoding reactions only marginally increased genome coverage in the sample from room D14 from 99.02% to 99.26%.

## Recovered genome sequences are from clade 19B may have originated from a single patient, or from multiple patients infected with similar viruses

To compare the near-complete genome sequences generated, we conducted phylogenetic analyses. We first determined that the pairwise identity between these two genomes was 93.8%, with several polymorphisms present. We conducted a phylogenetic analysis using NextStrain [39] to compare the sequences with other viruses detected through local subsampling in California and Sacramento County specifically. Both sequences were placed in clade 19B (Fig 5a), which were the first sequenced variants that circulated (along with 19A) in Asia early in the epidemic [40]. We included all publicly available samples sequenced from UCDMC in the phylogeny (Fig 5b). Both sequences clustered with UCDMC sample USA/CA-CZB-1145/2020, and notably these three samples clustered in an entirely different clade than the rest of the UCDMC samples, which were in clade 20C that arose in Europe. Thus, it appears likely these samples were derived from a single patient (or from multiple patients infected with similar viruses) from which USA/CA-CZB-1145/2020 originated.

## Conclusions

Eleven percent of samples collected at the UC Davis Medical Center in April 2020 were positive for SARS-CoV-2 whereas a larger follow-up experiment in August found only 2% of swabs positive, which is likely due to improved cleaning protocols and improved management of patient respiratory secretions. Near-complete genome sequences were amplified from two surfaces, suggesting the presence of viral genomes. However, in agreement with numerous other studies, no infectious virus was detectable from surfaces. Taken together, these findings suggest that while the virus on surfaces doesn't appear to be infectious, there is still a need for other mitigation measures to minimize transmission risk. Genome sequences from the positive samples at the first sampling point suggest that the environmental contamination was linked to a single lineage of virus, most likely from a single patient or from multiple patients with closely related infections. While interpreting our data, we need to take in consideration that

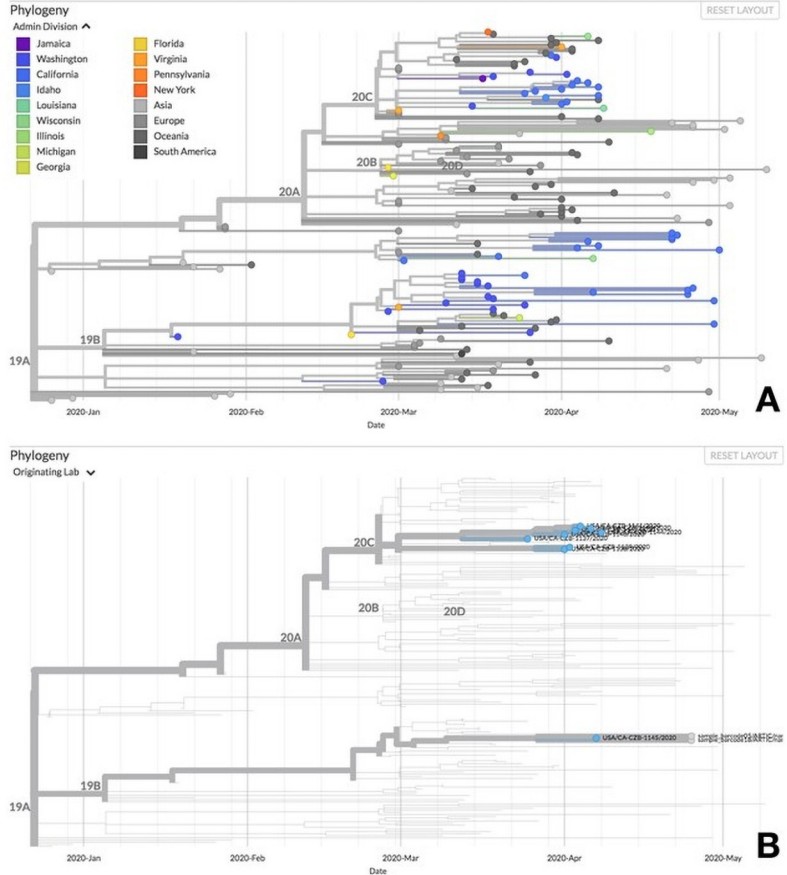

**Fig 5. Phylogenetic comparison of the SARS-CoV-2 sequences obtained from environmental swabs at UCDMC.**
**A**. Near-complete genomes obtained from environmental samples clustered in clade 19B. The phylogenetic tree was generated using the NextStrain protocol, and compares sequences to others amplified in Sacramento County in California. **B**. Environmental genome sequences may have originated from a single patient, or from multiple patients infected with similar viruses. All publicly available patient samples originating from UC Davis are shown as blue points at the tips of the phylogeny. Note that most sequences from UC Davis in this time period are members of the 20C clade, as opposed to the environmental sequences that are members of clade 19B together with sample USA/CA-CZB-1145/2020.

this study was conducted in just one medical center at a specific humidity, temperature, UV light, ventilation, occupancy, activity level and environmental context and at the time when the different variants of SARS-Co-V-2 that we see currently were not known. Importantly, we show here that viral sequences could be amplified from samples that were negative by qRT-PCR, highlighting the superior sensitivity of this technique and raising its potential suitability to identify SARS-CoV-2 variants from environmental samples.

## Supporting information

**S1 Table. Locations of samples positive for SARS-CoV-2 by qRT-PCR.** "U" is Undetermined (at 45 cycles of qRT-PCR). All patient rooms were occupied by known COVID-19 cases. The 1st wave was in the spring of 2020, and the second was in late summer 2020. (DOCX)

**S2 Table. Locations and qRT-PCR results for all samples collected.** Undetermined is at 45 cycles of qRT-PCR.
(DOCX)

**S3 Table. Sequencing information for 5 MinION runs, detailing number of raw reads generated and the amount retained at each step of the bioinformatics pipeline.**
(DOCX)

## Acknowledgments

The authors would like to thank Samantha Levy for help with transcribing data and collecting references, and the UC Davis Pulmonary and Critical Care Clinical Research Unit staff for collecting the clinical samples.

## Author Contributions

**Conceptualization:** David A. Coil, Timothy Albertson, Greg Brennan, Stuart H. Cohen, Satya Dandekar, Samuel L. Díaz-Muñoz, Jonathan A. Eisen, Tracey Goldstein, Maya Juarez, Brandt A. Robinson, Stefan Rothenburg, Christian Sandrock, Angela Haczku.

**Data curation:** David A. Coil, Shefali Banerjee, A. J. Campbell, Samuel L. Díaz-Muñoz, Ivy R. Jose, Maya Juarez, Brandt A. Robinson, Daniel G. Tompkins, Alexandre Tremeau-Bravard.

**Formal analysis:** David A. Coil, Shefali Banerjee, Greg Brennan, A. J. Campbell, Samuel L. Díaz-Muñoz, Tracey Goldstein, Ivy R. Jose, Stefan Rothenburg, Ana M. M. Stoian.

**Funding acquisition:** David A. Coil, Jonathan A. Eisen, Angela Haczku.

**Investigation:** Timothy Albertson, Greg Brennan, Samuel L. Díaz-Muñoz, Tracey Goldstein, Maya Juarez, Brandt A. Robinson, Christian Sandrock, Ana M. M. Stoian, Daniel G. Tompkins, Alexandre Tremeau-Bravard.

**Methodology:** Samuel L. Díaz-Muñoz, Maya Juarez, Stefan Rothenburg, Ana M. M. Stoian, Alexandre Tremeau-Bravard.

**Project administration:** David A. Coil, Timothy Albertson, Angela Haczku.

**Software:** A. J. Campbell, Samuel L. Díaz-Muñoz.

**Supervision:** David A. Coil, Timothy Albertson, Stuart H. Cohen, Satya Dandekar, Samuel L. Díaz-Muñoz, Jonathan A. Eisen, Tracey Goldstein, Stefan Rothenburg, Christian Sandrock, Angela Haczku.

**Validation:** Ivy R. Jose.

**Visualization:** Shefali Banerjee, Ana M. M. Stoian.

**Writing – original draft:** David A. Coil, Greg Brennan, Samuel L. Díaz-Muñoz, Stefan Rothenburg, Angela Haczku.

**Writing – review & editing:** David A. Coil, Timothy Albertson, Shefali Banerjee, Greg Brennan, A. J. Campbell, Stuart H. Cohen, Satya Dandekar, Samuel L. Díaz-Muñoz, Jonathan A. Eisen, Tracey Goldstein, Ivy R. Jose, Maya Juarez, Brandt A. Robinson, Stefan Rothenburg, Christian Sandrock, Ana M. M. Stoian, Daniel G. Tompkins, Alexandre Tremeau-Bravard, Angela Haczku.

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
