## [Decision Letter · Decision Letter 0]

17 May 2021

PONE-D-21-11199

SARS-CoV-2 detection and genomic sequencing from hospital surface samples collected at UC Davis

PLOS ONE

Dear Dr. Coil,

Thank you for submitting your manuscript to PLOS ONE. After careful consideration, we feel that it has merit but does not fully meet PLOS ONE’s publication criteria as it currently stands. Therefore, we invite you to submit a revised version of the manuscript that addresses the minor points raised during the review process.

We look forward to receiving your revised manuscript.

Kind regards,

Binod Kumar, PhD

Academic Editor

PLOS ONE

Journal Requirements:

Reviewers' comments:

Reviewer's Responses to Questions

**Comments to the Author**

1. Is the manuscript technically sound, and do the data support the conclusions?

Reviewer #1: Yes

Reviewer #2: Yes

2. Has the statistical analysis been performed appropriately and rigorously? 

Reviewer #1: N/A

Reviewer #2: Yes

3. Have the authors made all data underlying the findings in their manuscript fully available?

Reviewer #1: Yes

Reviewer #2: Yes

4. Is the manuscript presented in an intelligible fashion and written in standard English?

Reviewer #1: Yes

Reviewer #2: Yes

5. Review Comments to the Author

Reviewer #1: The Manuscript by Coil et al deals with an important issue in the current pandemic situation whether different hospital surfaces possess contagious and infective SARS-Cov2 viral particles. The study clearly showed that although viral genome is found in many surfaces in hospital setting but they are not contagious at all. The authors also identified that the important issue related to proper cleaning management helps to reduce RT-PCR positivity in different hospital surfaces. Using whole genome sequencing authors recovered a near-complete viral genome from various surface swabs.

Overall the work is impressive with few minor concerns listed below:

1. While constructing the phylogenetic tree, the authors claim that the samples where near-complete genome was recovered by sequencing were derived from single patient. This might also be possible that multiple patients were infected with similar virus since at the time of study only few variants/mutants were in USA/UCDMC. This statement needs to be corrected both in results section (line – 289) and conclusion (line 301-302) section. That might be more correct scientific way to discuss the issue.

2. There are couple of reports in others study also showed that there is minimal chance of viral infectivity from different surfaces. There are also several reports suggest that cleaning reduces viral particles on surfaces. Thus this manuscript lacks novelty in this perspective.

3. The conclusion section needs to be elaborated with reference to previous findings in this context and a brief necessary discussion will help readers to better understand the importance of this work.

4. Figure Legends section – Figure 5 heading needs to be bold character.

Reviewer #2: The manuscript by Coil et. al. aims to study the surface swab of SARS-COV2- RNA and determine its applicability in sequencing studies. Overall, this is a well written manuscript which looks at an extremely relevant topic and will add to the existing knowledge about the COVID-19 pandemic. Furthermore, this study has the potential to be informative in cases where a new cleaning method is being tested in places such as long-term senior care facilities as we all to evaluate the effectiveness of different control measures. Although the study provides evidence of low likelihood that hospital surface samples (with proper cleaning method in place) contains low likelihood of active transmission the article can include in the conclusion that the results should not encourage to replace the existing public health measures of keeping good hygiene and physical distancing to prevent transmission. Additionally, the study should mention in the discussion that the conclusion should be considered keeping the following in mind that it was conducted in just one medical center at a specific humidity, temperature, UV light, ventilation, occupancy, activity level and environmental context and importantly at a time when the different variants of SARS-COV2 that we see currently were not known. Please also mention the for which gene the RT-PCR were conducted in the main text. Overall, this is a relevant and well written manuscript.

6. PLOS authors have the option to publish the peer review history of their article (what does this mean?). If published, this will include your full peer review and any attached files.

Reviewer #1: **Yes: **Rupkatha Mukhopadhyay

Reviewer #2: No

---

## [Decision Letter · Decision Letter 1]

9 Jun 2021

SARS-CoV-2 detection and genomic sequencing from hospital surface samples collected at UC Davis

PONE-D-21-11199R1

Dear Dr. Coil,

We’re pleased to inform you that your manuscript has been judged scientifically suitable for publication and will be formally accepted for publication once it meets all outstanding technical requirements.

Kind regards,

Binod Kumar, PhD

Academic Editor

PLOS ONE

Additional Editor Comments (optional):

Reviewers' comments:

Reviewer's Responses to Questions

**Comments to the Author**

1. If the authors have adequately addressed your comments raised in a previous round of review and you feel that this manuscript is now acceptable for publication, you may indicate that here to bypass the “Comments to the Author” section, enter your conflict of interest statement in the “Confidential to Editor” section, and submit your "Accept" recommendation.

Reviewer #1: All comments have been addressed

Reviewer #2: (No Response)

2. Is the manuscript technically sound, and do the data support the conclusions?

Reviewer #1: Yes

Reviewer #2: (No Response)

3. Has the statistical analysis been performed appropriately and rigorously? 

Reviewer #1: N/A

Reviewer #2: (No Response)

4. Have the authors made all data underlying the findings in their manuscript fully available?

Reviewer #1: Yes

Reviewer #2: (No Response)

5. Is the manuscript presented in an intelligible fashion and written in standard English?

Reviewer #1: Yes

Reviewer #2: (No Response)

6. Review Comments to the Author

Reviewer #1: The manuscript by Coil et al., addresses all the points that were raised and their reviewed version of manuscript improved substantially. Overall the study is important for the identification of the source of contagion of SARS-CoV at the hospital setting in the current pandemic scenario.

Reviewer #2: (No Response)

7. PLOS authors have the option to publish the peer review history of their article (what does this mean?). If published, this will include your full peer review and any attached files.

Reviewer #1: **Yes: **Rupkatha Mukhopadhyay

Reviewer #2: No

---

## [Editor Report · Acceptance letter]

14 Jun 2021

PONE-D-21-11199R1 

SARS-CoV-2 detection and genomic sequencing from hospital surface samples collected at UC Davis 

Dear Dr. Coil:

I'm pleased to inform you that your manuscript has been deemed suitable for publication in PLOS ONE. Congratulations! Your manuscript is now with our production department. 

Kind regards, 

on behalf of

Dr. Binod Kumar 

Academic Editor

PLOS ONE